# Joint Learning of Vessel Segmentation and Artery/Vein Classification with Post-processing

**Liangzhi Li**                       LI@IDS.OSAKA-U.AC.JP
**Manisha Verma**                 MVERMA@IDS.OSAKA-U.AC.JP
**Yuta Nakashima**                N-YUTA@IDS.OSAKA-U.AC.JP
**Ryo Kawasaki**         RYO.KAWASAKI@OPHTHAL.MED.OSAKA-U.AC.JP
**Hajime Nagahara**              NAGAHARA@IDS.OSAKA-U.AC.JP
*Osaka University, 1-1 Yamadaoka, Suita, Osaka, Japan 565-0871*

## Abstract

Retinal imaging serves as a valuable tool for diagnosis of various diseases. However, reading retinal images is a difficult and time-consuming task even for experienced specialists. The fundamental step towards automated retinal image analysis is vessel segmentation and artery/vein classification, which provide various information on potential disorders. To improve the performance of the existing automated methods for retinal image analysis, we propose a two-step vessel classification. We adopt a UNet-based model, SeqNet, to accurately segment vessels from the background and make prediction on the vessel type. Our model does segmentation and classification sequentially, which alleviates the problem of label distribution bias and facilitates training. To further refine classification results, we post-process them considering the structural information among vessels to propagate highly confident prediction to surrounding vessels. Our experiments show that our method improves AUC to 0.98 for segmentation and the accuracy to 0.92 in classification over DRIVE dataset.

**Keywords:** Medical imaging, retina images, vessel segmentation, vessel classification, deep learning, computer vision.

## 1. Introduction

Retinal imaging is the only feasible way to directly inspect the vessels and the central nervous system in the human body *in vivo*, which can give us informative signs and indications on possible disorders. Fundoscopy has thus become an important method and the routing examination to help diagnosis of many diseases, including diabetes, hypertension, arterial hardening, and so forth (Chatziralli et al., 2012). Fundoscopy is easy to operate, quick, accurate, and relatively low in cost. Medical doctors, not only ophthalmologists, are considering a wider use of fundoscopy.

However, similarly to other types of medical images, retina images exhibit high complexity and huge diversity (Jin et al., 2019). Sufficiently trained specialists are required to handle ever-increasing requests to read such images. Moreover, reading retinal images by specialists can potentially be error-prone under this highly demanded circumstance. To that end, computer-aided diagnosis can be a promising technical break-through that automatically analyzes such retina images.

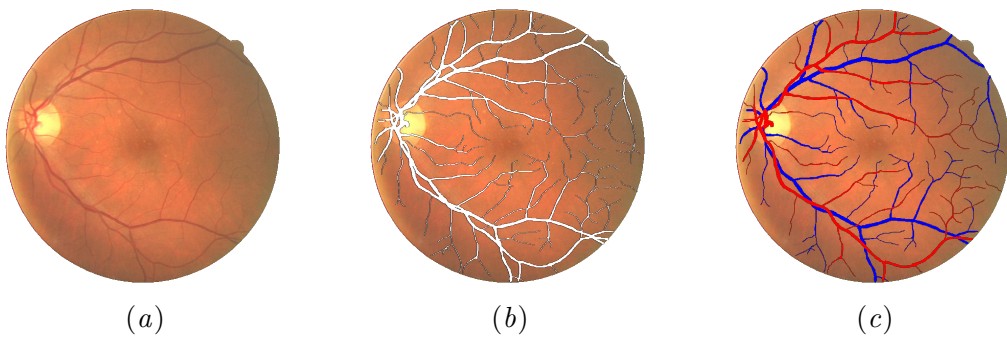

$(a)$                  $(b)$                  $(c)$

Figure 1: An example retina image from a public dataset (Staal et al., 2004; Hu et al., 2013). (a) Raw image. (b) Vessel segmentation. (c) Artery (red) / Vein (blue) classification.

Various high-level tasks of retinal image analysis, such as the calculation of central artery equivalent, central vein equivalent, artery-to-vein diameter ratio (Huang et al., 2018b), as well as the detection of retinal artery occlusion and retinal vein occlusion (Woo et al., 2016), which can reveal risks of stroke, cerebral atrophy, cognitive decline, and myocardial infarct, etc., are built on top of vessel segmentation and artery/vein (A/V) classification. A vast amount of research efforts have been made for both components. For vessel segmentation, most of the earliest attempts are based on the local information of retinal images (Cheng et al., 2014; Roychowdhury et al., 2015), including intensity, color, some hand-crafted features, etc. In recent years, UNet (Ronneberger et al., 2015)-based segmentation models become more popular (Kim et al., 2017; Yan et al., 2018). As for A/V classification, a classic approach is applied to segmented vessels in retinal images (Huang et al., 2018a), where some structural prior on vessels has been leveraged for better performance (Alam et al., 2018; Srinidhi et al., 2019). Deep models are also explored and achieved the state-of-the-art performance (Meyer et al., 2018). Meanwhile, lack of large-scale labeled datasets motivates data augmentation with generative adversarial networks (Costa et al., 2018).

Although many approaches have been proposed in this area, their performances are not satisfactory yet. This is because the retina images are usually complicated and full of noises. It is hard to extract all vessels, including minor ones, while not introducing too many false vessel pixels. Moreover, the available training data are very limited. In most of the public datasets, the number of retina images for training is no more than 20. Furthermore, things become more difficult when we need to classify the vessels into artery or vein, because this further increases the unbalance between the numbers of pixels on artery or vein vessels and the number of background (non-vessel) pixels.

In this paper, we propose a method for automatically analyzing retinal images, such as the one in Fig. 1. Our method consists of two components: (i) A neural model, coined SeqNet, that segments vessels and classifies each pixel into artery and vein, and (ii) post-processing to refine initial classification by SeqNet. The main idea behind our neural model is to jointly training the model, but yet segmentation and classification streams are sequen-

tial rather than simultaneous, as shown in Fig. 2. The segmentation stream only cares about vessel extraction. Meanwhile, the classification stream utilizes segmentation results to immunize itself against cluttered backgrounds in input images. The existing methods that simultaneously do segmentation and classification suffer from the severe bias in label distributions since background pixels are dominant in retinal images. We remedy this imbalance by our sequential model, dividing the task into the background/vessel classification (i.e. segmentation) task and artery/vein classification task, where we employ the state-of-the-art model (Li et al., 2020) for the segmentation stream.

There may still be some errors in classification results. This is because fully convolutional network-like models (such as UNet-based ones (Meyer et al., 2018; Hemelings et al., 2019; Galdran et al., 2019)), or more generally convolution operations, are more suitable to extract local features than handling global context. Hence all UNet-based models' prediction performances depend on local cues, such as color and contrast, rather than the structure of the whole vessel system. This locality leads to many minor errors, as shown in Fig. 6(a) and (b).

We thus incorporate the global context, i.e., the structure of the vessel system, into our method via post-processing for further improving the performance. We divide extracted vessels into many small segments and unifying the pixel-level predictions in each of them into a single prediction, called intra-segment label unification. We also propose a new strategy called inter-segment prediction propagation (PP). This strategy can further refine classification results among neighboring segments by propagating predictions to neighboring segments with judging whether they are connected with each other or just crossed two different vessels.

Our main contribution is three-fold:

- We design a joint segmentation and classification model based on the UNet architecture (Ronneberger et al., 2015), which sequentially handles respective tasks to balance the label distributions for better training.

- We propose to post-process classification results for refining them by leveraging global information, called intra-segment label unification and inter-segment prediction propagation, which smooths each pixel's label along the vessel system's structure.

- We experimentally demonstrate that our method, including SeqNet and the post-processing, achieves the state-of-the-art performance over two public datasets. The code is available here[1].

## 2. Methodology

Our method consists of SeqNet (Fig. 2) for initial segmentation/classification and PP for refinement. Following sections details these two components.

### 2.1. SeqNet

Some existing methods for A/V classification actually formulate the problem as a ternary classification task, where each pixel is labeled as either artery, vein, or background. This

---

1. https://github.com/conscienceli/SeqNet

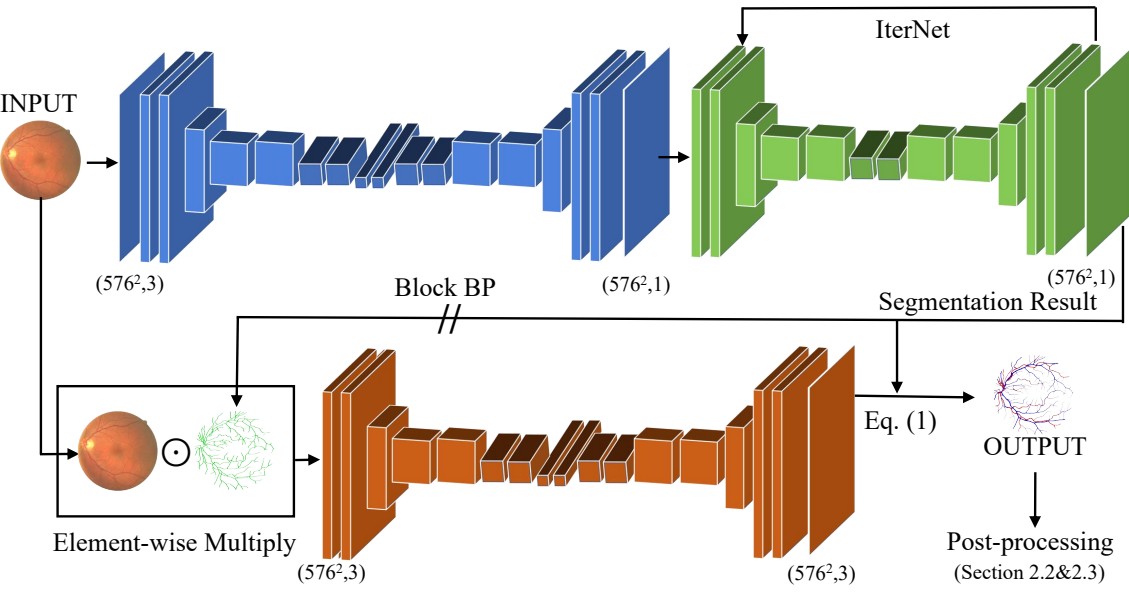

Figure 2: The network architecture of SeqNet.

can deteriorate the performance by imposing further imbalance among the labels, i.e., there are much more background labels than artery/vein labels. Most state-of-the-art models actually suffer from a poor segmentation ability, which is discussed in Section 3. Unlike these methods, SeqNet sequentially applies segmentation into vessel/background and classification into A/V in a single network. Yet, training is done jointly.

As shown in Fig. 2, SeqNet mainly consists of two streams (the upper stream with the blue and green blocks and the lower stream with the orange block). The upper stream is for segmentation. We adopt IterNet (Li et al., 2020), which iteratively refines the segmentation results by smaller UNets (the green block in Fig. 2) after initial segmentation by the blue block. The state-of-the-art performance has been achieved with this model over the mainstream datasets (Staal et al., 2004; Tang et al., 2011). In SeqNet, the green block is repeated three times, following the original implementation in (Li et al., 2020). Both two streams use separate cross entropy losses and are trained jointly with a batch size of 16. For the target, IterNet uses the segmentation labels while the classification part uses the A/V labels. Adam (Kingma and Ba, 2014) is used as the optimizer with a learning rate of 0.001.

With input retinal image $\mathbf{x} \in \mathbb{R}^{W \times H}$ and refined vessel map $\mathbf{v} \in [0, 1]^{W \times H}$ by IterNet, where $W = 576$ and $H = 576$ are the width and height the input image and vessel map, we apply another full-size UNet block, which is shown in orange in Fig. 2, to classify each pixel into artery/vein. The possible output labels are *background*, *artery*, and *vein*. We mask *background* pixels in input image $\mathbf{x}$ by

$$\mathbf{x}' = \mathbf{x} \odot \mathbf{v}, \tag{1}$$

where $\odot$ is the element-wise multiplication. This masking reduces the complexity of the input retinal image, so that the classification stream can fully focus on finding the differences in color, thickness, shape, etc., among the vessels. We put a block layer before the element-wise multiplication to prevent back-propagation from the classification stream to the segmentation stream, so that each steam can be responsible to the respective task and can be trained in a multi-task manner.

The output from the classification stream is merged with the segmentation result. Let $\mathbf{o}_l \in [0, 1]^{W \times H}$, where $l \in \{background, artery, vein\}$ denote the softmax output of the classification stream.

## 2.2. Intra-segment Label Unification

There are mainly two types errors in classification results: The first one is inconsistency along one single vessel, i.e., both *artery* and *vein* labels appear in a vessel, as shown in Fig. 7, because the underlying convolutional network does not count the structure of the vessel system, making decisions mainly based on local features, such as color and shape. These local features can be easily influenced by environmental factors, e.g., illumination and the retinal camera used. The second type of errors is mixed-up prediction that happens mostly near the crossing and branching points, as shown in Fig. 8, because local features corresponding to both vessel types may be observed. To remedy these two kinds of errors, we design a post-processing algorithm, namely, *intra-segment label unification* for the label inconsistency problem and *inter-segment prediction propagation* for the mixed-up prediction problem.

Intra-segment label unification firstly generates a binary image $\mathbf{p}$ of detected vessels from SeqNet's output $\mathbf{v}$ by:
$$p_k = \mathbb{1}\{v_k > \theta\}, \tag{2}$$
where $p_k$ and $v_k$ are the $k$-th pixels in $\mathbf{p}$ and $\mathbf{v}$, respectively; $\theta$ is a predefined threshold. We then extract binary skeletons using a multiple-threshold method introduced in Appendix A, as shown in Fig. 3(a). We detect all *key-points*, which includes the crossing points between vessels and the terminal points (i,.e., start and end points) of vessels (Fig. 3(b)). Crossing points are detected by looking for vessel pixels on the skeleton image that have more than two neighbors, while terminal points only have no more than one neighbor. Skeletal pixels between connected key-points are extracted as a *segment* as in Fig. 3(c).

Let $S = \{S_i | i = 1, \ldots, N\}$ be the set of all $N$ segments extracted from $\mathbf{p}$, where $S_i$ is the set of pixels in segment $i$. We compute the confidence $c_i^l$ that segment $S_i$ belongs to $l$ in $\{artery, vein\}$ by
$$c_{li} = \sum_{k \in S_i} (o_{artery,k} - o_{vein,k}), \tag{3}$$
where $o_{lk}$ is the value in $\mathbf{o}_l$ corresponding to pixel $k$. $c_{li}$ can be viewed as unified label confidence of $S_i$ corresponding to $l$, where actual prediction can be done by comparing $c_{li}$'s, i.e., $S_i$ is *artery* if $c_{artery,i} > c_{vein,i}$ and *vein* otherwise.

## 2.3. Inter-segment Prediction Propagation

To address errors around crossing and branching points, we introduce additional post-processing, coined inter-segment prediction propagation, in which the label of a segment is

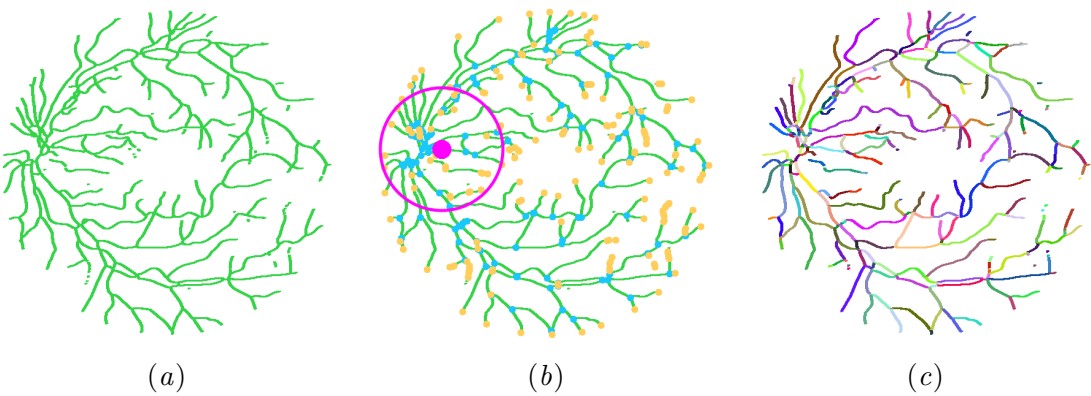

$(a)$ $(b)$ $(c)$

Figure 3: An illustrative example of intra-segment label unification. (a) Extracted vessel skeleton. (b) Detected key-points (magenta dot for the cup center and circle for the cup area; blue for crossing points; yellow for terminal points). (c) Extracted segments.

propagated to its connected segments. This is based on the observation that classification failures usually come with a low confidences on their labels and that they can be corrected by the influences from their connecting segments with high confidence. Propagation should happen depending on the similarity between connecting segments based on their shapes, directions, etc. If two segments share similar shapes, are located nearby, and flows in similar directions, it is highly possible that they belong to the same vessel. Therefore, the influence between these segments should be strong.

Based on this observation, we update confidence $c_{li}$ of segment $S_i$ according to the following rule:

$$c_{li} \leftarrow c_{li} + \epsilon_{ij} c_{lj} \tag{4}$$

where $j$ is the index of segment connected to $i$. $\epsilon$ is the coefficient to determine the influence of $S_j$ to $S_i$, given by

$$\epsilon_{ij} = A_{ij} L_{ij} T_{ij} D_{ij} \tag{5}$$

Let $\mathbf{u}_i$ be the unit tangent vector of $S_i$ at a certain key-point, which is computed using the key-point pixel position $\mathbf{p}_{i1}$ and the position $\mathbf{p}_{i5}$ of the fifth pixel along the skeleton, i.e., $\mathbf{u}_i = (\mathbf{p}_{i5} - \mathbf{p}_{i0})/\|\mathbf{p}_{i5} - \mathbf{p}_{i0}\|$. $A$ involves the angle between $\mathbf{u}_i$ and $\mathbf{u}_j$, defined as

$$A_{ij} = F_{\mathrm{A}}(|\alpha(\mathbf{u}_i, \mathbf{u}_j) - 180|) \tag{6}$$

where $\alpha(\mathbf{u}_i, \mathbf{u}_j)$ is the angle formed by segments $\mathbf{u}_i$ and $\mathbf{u}_j$ and $F_{\mathrm{A}}$ is given by

$$F_{\mathrm{A}}(x) = \frac{(x - m_{\mathrm{A}})^2}{m_{\mathrm{A}}^2}, \tag{7}$$

where $m_{\mathrm{A}}$ is the pre-defined maximum value decided by observing the vessel systems on the training images. This function serves as normalization of $x$ into $[0, 1]$. $A_{ij}$ gives 1 if the tangent vectors are in the opposite directions (i.e., $\alpha(\mathbf{u}_i, \mathbf{u}_j)$ gives 180 degree).

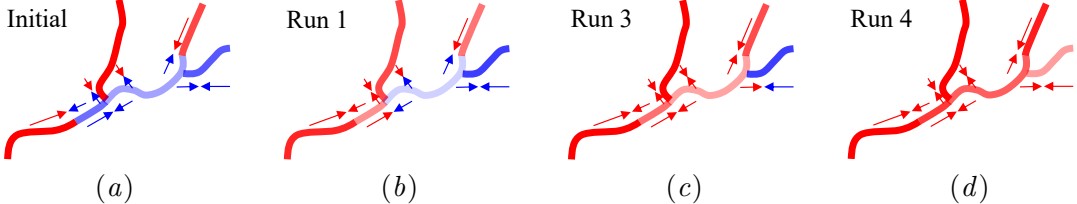

Figure 4: An illustrative example of prediction propagation. (a) Initial prediction with several errors. (b)–(d) Resulting predictions after individual iterations.

$L$ handles a potential missing connection between two segments, which is defined as

$$L_{ij} = F_{\mathrm{L}}(\alpha(\mathbf{u}_i, \mathbf{w}_{ij})), \tag{8}$$

where $\mathbf{w}_{ij}$ is a unit vector from $S_i$'s key-point to $S_j$'s, and the angle computed by $\alpha$ is normalized by $F_{\mathrm{L}}$ in the same way as Eq. (7). $L_{ij}$ gives a value close to 1 if one of $S_j$'s key-point is on the line described by $\mathbf{w}_{ij}$.

Thickness of vessels can also be a informative cue to retrieve connecting vessels since they share a similar thickness when they are connected to each other. We encode this by $T_{ij}$, defined as

$$T_{ij} = F_{\mathrm{T}}(\beta(S_i, S_j)) \tag{9}$$

where $\beta(S_i, S_j)$ gives the difference of mean thickness of $S_i$ and $S_j$, computed along the skeleton pixels. $D_{ij}$ gives a small value if $S_i$ and $S_j$ are far from each other. We defined this as

$$D_{ij} = F_{\mathrm{D}}(\|\mathbf{p}_{i0} - \mathbf{p}_{j0}\|). \tag{10}$$

Both $F_{\mathrm{T}}$ and $F_{\mathrm{D}}$ are defined in the same way as Eq. (7).

We apply this update rule to all extracted segments. The detailed algorithm is presented in Algorithm 1 in Appendix. The label confidence $c_i$ evolves as shown in Fig. 4. We can see that several iterations correct the predicted labels. Note that a segment has two end points, while $A_{ij}$, $L_{ij}$, and $D_{ij}$ involve a single end point in each of segments $S_i$ and $S_j$. We update the confidence for all four combinations of end points.

This propagation process is not allowed to change the segments in the cup area, which is indicated by the magenta circle in Fig. 3(b). This is because vessels in this area are too dense and hard to analyze their relationships, i.e., which segments are actually connected together and which segments are merely crossing, etc. Also, higher brightness in the cup area results in many segmentation failures, which may lead to the failure of PP.

## 3. Performance Evaluation

We use two popular public datasets, namely DRIVE (Staal et al., 2004), and the artery/vein labels from (Hu et al., 2013), as well as LES-AV (Orlando et al., 2018), to evaluate our method. We compare our method with two recent methods, *i.e.*, uncertainty-aware (UA)

Table 1: Performance evaluation on DRIVE dataset.

| Methods | Full Image | Center | | Center$_{\geq 2\,px}$ | | Vessel |
|---|---|---|---|---|---|---|
| | | Acc. | F1 | Acc. | F1 | |
| UA (Galdran et al., 2019) | 0.966 | 0.888 | 0.888 | 0.923 | 0.923 | 0.741 |
| FCN (Hemelings et al., 2019) | - | - | - | 0.940 | - | - |
| SeqNet *w.o.* post-processing | **0.967** | 0.914 | 0.914 | 0.946 | 0.946 | 0.774 |
| SeqNet *w.* post-processing | **0.967** | **0.919** | **0.919** | **0.953** | **0.953** | **0.778** |

Table 2: Performance evaluation on LES-AV dataset.

| Methods | Full Image | Center | | Center$_{\geq 2\,px}$ | | Vessel |
|---|---|---|---|---|---|---|
| | | Acc. | F1 | Acc. | F1 | |
| SeqNet *w.o.* post-processing | **0.978** | 0.858 | 0.858 | 0.916 | 0.916 | 0.776 |
| SeqNet *w.* post-processing | **0.978** | **0.874** | **0.874** | **0.930** | **0.930** | **0.785** |

(Galdran et al., 2019) and fully convolutional network (FCN) (Hemelings et al., 2019), on the DRIVE dataset.

One problem is that existing methods use different evaluation strategies. Although most of them use accuracy as the performance metric, but usually with different pixel masks, including the whole image, the discovered vessel pixels, the ground-truth vessel pixels, the major vessel pixels, etc. To remove the barrier of reproducing and testing A/V classification methods, we adopt a newly-proposed evaluation procedure (Hemelings et al., 2019) which includes a series of pixel masks, such as full image, center-line of discovered vessels, center-line of major discovered vessels (width$_{\geq 2\,px}$), the amount of discovered vessels, etc.

Among these results shown in Table. 1 and Table. 2, we can see that our method achieves a better AUC value than other models, as our model avoids deterioration of the segmentation performance due to isolation of segmentation and classification. Also, our full method (SeqNet & LU & PP) shows higher accuracy on both datasets.

## 4. Conclusion

In this paper, we propose SeqNet for accurate vessel segmentation and artery/vein classification in retinal images, together with a post-processing algorithm. SeqNet sequentially does segmentation and classification but not simultaneously, which may deteriorate the segmentation performance due to the problem of imbalanced label distribution. Our post-processing algorithm then corrects classification results by propagating highly confident labels to their surrounding vessels segments. Experimental results showed that our method is effective and can achieve the state-of-the-art performance on two public datasets.

## Acknowledgments

This work was supported by Council for Science, Technology and Innovation (CSTI), cross-ministerial Strategic Innovation Promotion Program (SIP), "Innovative AI Hospital Sys-

tem" (Funding Agency: National Institute of Biomedical Innovation, Health and Nutrition (NIBIOHN)). This work was also supported by JSPS KAKENHI Grant Number 19K10662.

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

## Appendix A. Multiple Thresholds in Segments Extraction

In order to propagate the influence correctly, we have to extract the vessel segments accurately. Otherwise, the vessel map may be erroneous, resulting in unreasonable propagation, as shown in Fig. 5(a). Due to a missing important segment, a wrong label is propagated to the segment on the right hand side. Therefore, we should make several different binary skeleton with different thresholds and combine them into a complete vessel map. This is also detailed in Algorithm 1.

## Appendix B. Example Results of Intra-Segment Label Unification

Fig. 6(a) shows the direct output from the classification stream, in which we can see many prediction errors. Figs. 6(b) and (c) are the results of vessel skeleton extraction and label unification, respectively, where most label inconsistency in a single vessel segment have been resolved.

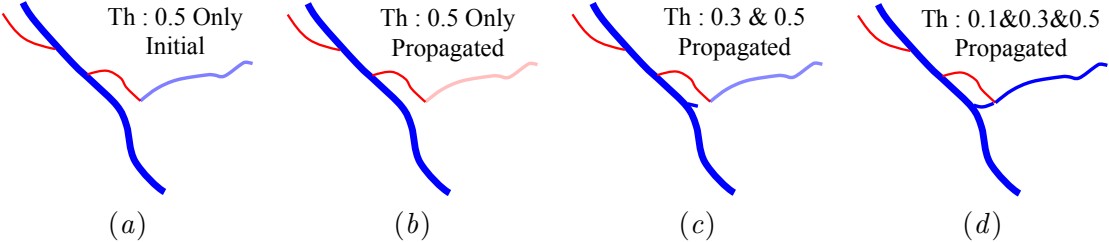

Figure 5: Multiple thresholds and the propagation results.

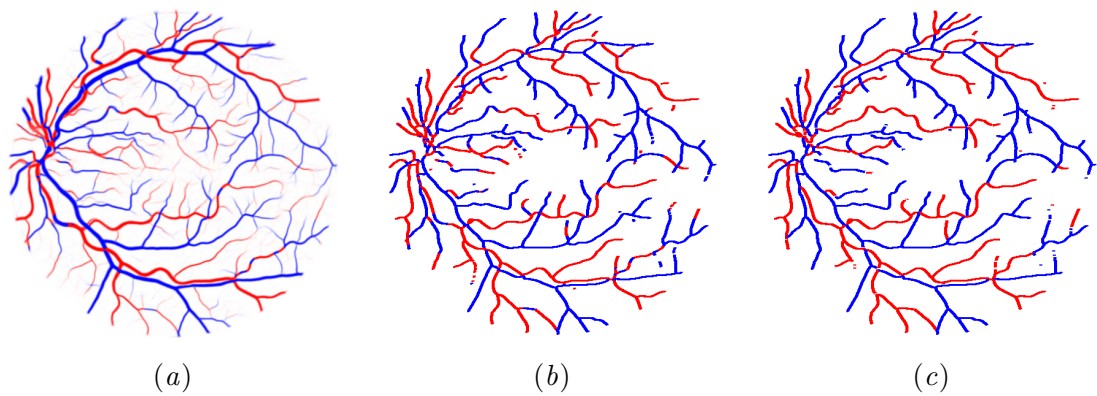

Figure 6: Example of label unification. (a) The initial prediction by SeqNet. (b) Vessel skeleton extracted from initial prediction. (c) Label unification result.

## Appendix C. Common Prediction Errors

Figs. 7 and 8 respectively show two common errors in classification, i.e., inconsistency along one single vessel segment and mixed-up prediction that happens around the crossing and branching points in most cases.

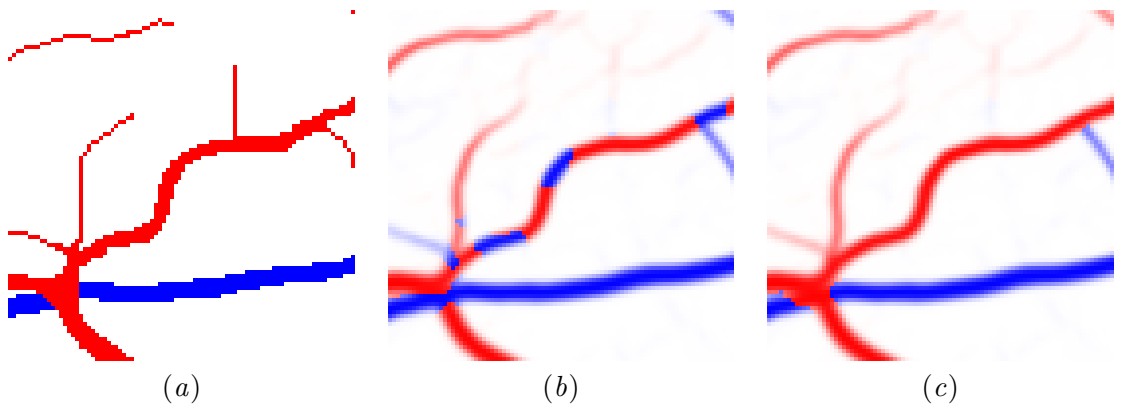

$(a)$ $(b)$ $(c)$

Figure 7: Prediction errors happened along a vessel segment. (a) Ground-truth labels. (b) Initial prediction by SeqNet. (c) Post-processed result.

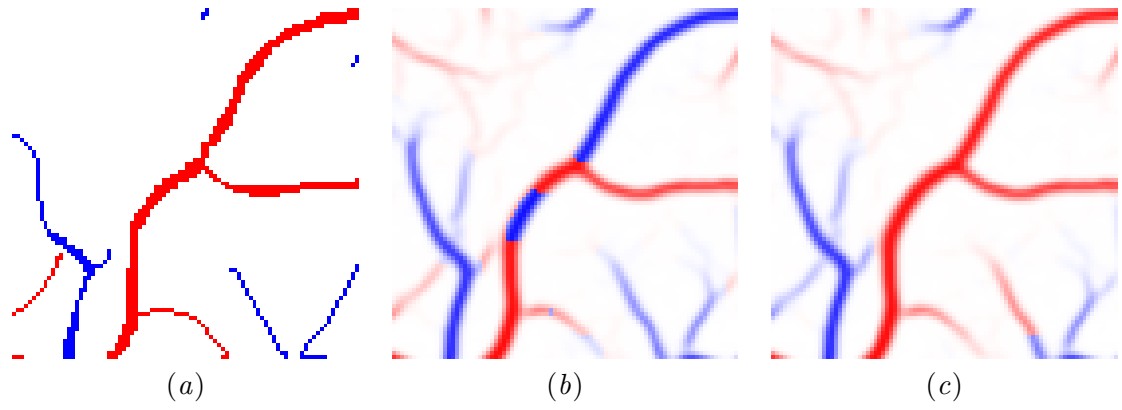

$(a)$ $(b)$ $(c)$

Figure 8: Prediction errors happened around crossing or branching points. (a) Ground-truth labels. (b) Initial Prediction by SeqNet. (c) Post-processed result.

## Appendix D. Post-Processing Algorithm

We detail the proposed post-processing in Algorithm 1, including multiple thresholds fusion, segment extraction, label unification, and prediction propagation.

The thresholds we select in our implementation are 0.5, 0.3, and 0.1. They are in a descending order because the higher threshold can result in a skeleton in higher confidence by focusing more on major vessels, while the smaller thresholds covers minor vessels.

As introduced in Section 2.2, label unification is based on the confidence associated with each segment, which is actually the sum of the prediction confidence of pixels in that segment. The confidence value is also used in PP, which may need several iterations for a better result. In our experiment, the number of iterations is set to 5.

---

**Algorithm 1:** Segment extraction, label unification, and prediction propagation.

---

**Input:** Initial prediction result $P = \{P_1, P_2, ..., P_n\}$
**Output:** Refined prediction result $P' = \{P'_1, P'_2, ..., P'_n\}$
```
/* Start searching segments in the vessel map                        */
```
segments ← None;
**for** $tr$ $in$ $[0.5, 0.3, 0.1]$ **do**
    BS ← Skeletonize(Binarify($P$, threshold=tr));
    keypoints ← FindEndPoints(BS) + FindCrossingPoints(BS);
    segments ← segments + FindSegments(keypoints);
**end**
```
/* Start unify the segments                                          */
```
**for** $S$ $in$ $segments$ **do**
    $t^S$ ← CalculateTotalConfidence($S$) ;           // using Eq. 3
    UnifyResultAlongOneSegment($S$);
**end**
```
/* Start prediction propgation                                       */
```
count ← 0;
**while** $count < 5$ **do**
    **for** $S$ $in$ $segments$ **do**
        $t^S$ ← UpdateConfidence($S$, segments) ;     // using Eq. 4,5
        ChangeSegmentCategory($S$, $t^S$);
    **end**
    count ← count +1;
**end**

---

## Appendix E. Example Prediction Results

Figs, 9 shows an example result on the DRIVE dataset.

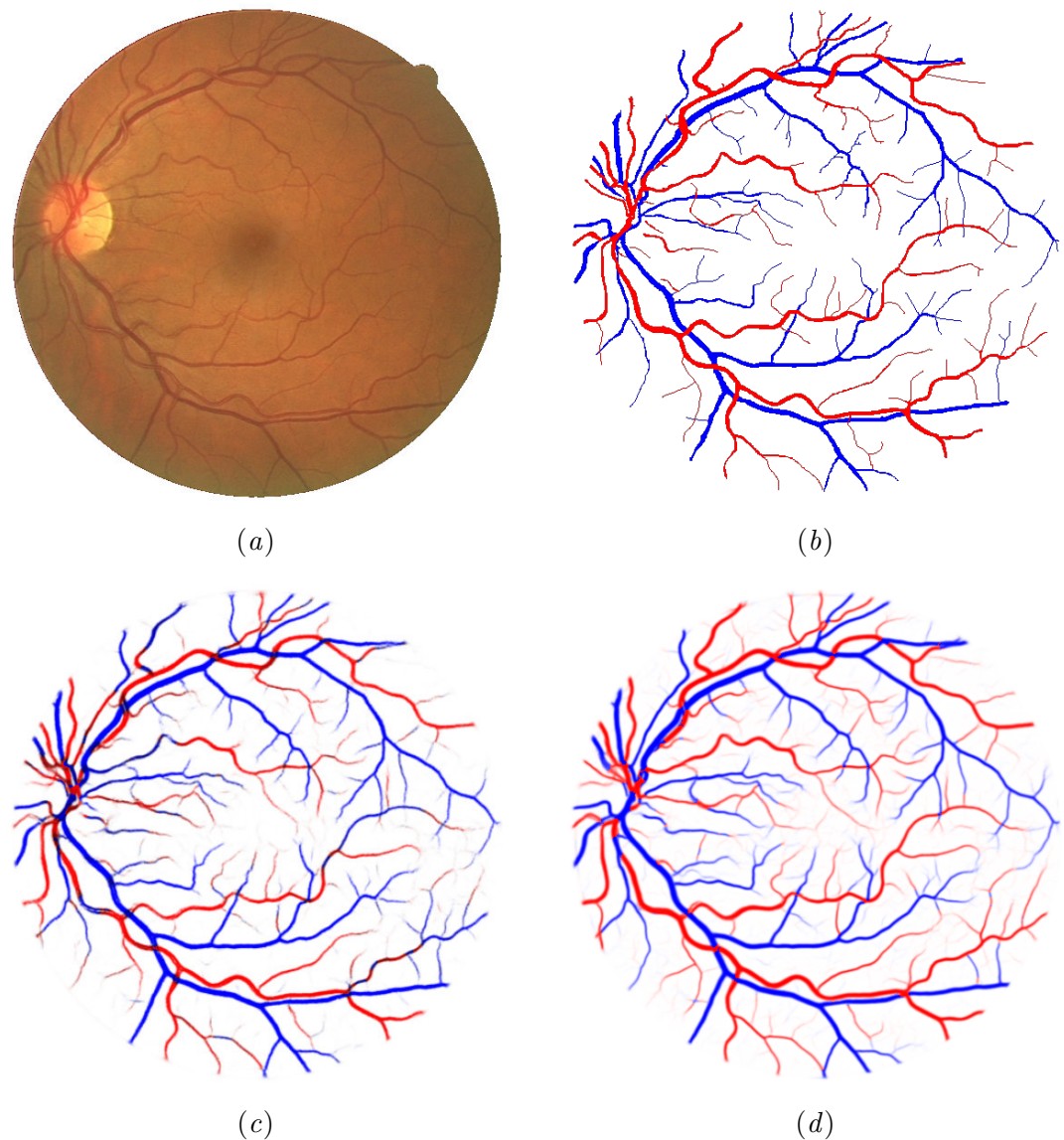

$(a)$ $(b)$

$(c)$ $(d)$

Figure 9: Prediction results for a single retinal image from the DRIVE dataset. (a) The input image. (b) The corresponding ground-truth labels. (c) The output from the uncert-aware method (Galdran et al., 2019). (d) The output from our method.

