# OpenReview forum: "Joint Learning of Vessel Segmentation and Artery/Vein Classification with Post-processing"
_MIDL.io/2020/Conference — MIDL 2020_

### Official Review · AnonReviewer4 · 2020-03-06
**Nice work, well described**

**Rating:** 4
**Confidence:** 4
**Recommendation:** Poster

**Summary:**

The authors describe an approach to segment the vessels in retinal images and additionally perform artery/vein separation. The method consists of an initial (U-net-like) segmentation, which is subsequently refined with an IterNet. The resulting final segmentation is fed into a netwerk that does the artery/vein labeling (pixel-wise). These classification results are subsequently improved in an (ad-hoc) post-processing step. In this processing step, a segment-wise majority-voting is applied, using either only the pixels in the segment, or also including a weighted contribution of neighboring segments. In the latter case, the weights are determined on various geometric properties that quantify how likely the two segments belong to the same vessel.

**Strengths:**


The manuscript is well written, a pleasure to read, and the methods are clearly explained. Whereas the Deep Learning part seems to be a straightforward utilization of existing techniques, I appreciate the combination with traditional approaches to include more global information in the processing, thus addressing the more local nature of U-net like approaches.

The authors apply there method on two existing databases, and demonstrate (a minor) improvement in performance. When assessing the DL part only, the results are inline with other approaches; the results demonstrates that the added post-processing is indeed able to improve the performance, bringing it slightly above other approaches.

**Weaknesses:**

Some issues could have been addressed. First, from the table I conclude that some approaches have a different performance over the two datasets, whereas the results of the authors approach seems to perform similar. I presume this is related to the training data (including images from both sets?). Information on the exact training, and possibly some discussion on this, may further improve the manuscript.

Whereas the results seem to improve consistently, the differences are minor. It would be relevant to 1) check whether these differences are statistically significant (if possible), and 2) discuss these improvements in the context of the clinical applications: how much would the patient (or physician) be off with the improved classification results.

In addition, add the value of all parameters (such as m_A) that were used in the final method. Similarly, if the networks used are not identical to the ones that the authors referred to, please specify the exact configuration (depth, initial nr. of features, drop-out, etc.)

**Justification Of Rating:**

Nice, well written manuscript with interesting method that combines Deep Learning with a 'traditional' post-processing approach.

Method is assessed on two known databases, and performs (slightly) better.

**Paper Type:**

both

**Questions To Address In The Rebuttal:**

Add detailed information on the training process.

Check whether the improvement is statistically significant (if this would be possible)

Add values of parameters used, and mention any deviations on network implementations with respect to the original works, if relevant.

**Special Issue:**

yes

---

> ### Author Response · Authors · 2020-03-27
> **Thanks for the comments**
>
> Thank you so much for your thoughtful review. We have summarized the changes in a top-level comment called "Summary of Changes in New Version".
>
> Check whether the improvement is statistically significant (if this would be possible)
> -> We keep trying to add statistical tests.

---

> > ### Author Response · Authors · 2020-04-04
> > **Other issues**
> >
> > 1.	Some issues could have been addressed. First, from the table I conclude that some approaches have a different performance over the two datasets, whereas the results of the authors approach seems to perform similar. I presume this is related to the training data (including images from both sets?). Information on the exact training, and possibly some discussion on this, may further improve the manuscript.
> >
> > In this revision, we redesign the experiments part, using new datasets and a new evaluation standard, as recommended by reviewer 2. The accuracy result is 0.953 (DRIVE) and 0.930 (LES-AV). We think these results show some differences between two datasets (refer to section (c) of “Summary of Changes” for the results using other metrics). In addition, we did not use any training materials shared between these two datasets.
> >
> > 2.	Whereas the results seem to improve consistently, the differences are minor. It would be relevant to 1) check whether these differences are statistically significant (if possible), and 2) discuss these improvements in the context of the clinical applications: how much would the patient (or physician) be off with the improved classification results.
> >
> > We rerun several times of the model and post-processing method and can observe a similar improvement on the classification performance. We will try to add the relevant results into the paper, but this may need some more time (all methods on all datasets should be rerun for statistical results). Regarding the clinical meaning of the improvement, we indeed have some work using the classification results for arteriolosclerosis diagnosis. Generally, if we can have better classification results, this diagnosis procedure will be smoother and much more accurate. We will add these discussions in the paper.
> >
> >
> > 3.	In addition, add the value of all parameters (such as m_A) that were used in the final method. Similarly, if the networks used are not identical to the ones that the authors referred to, please specify the exact configuration (depth, initial nr. of features, drop-out, etc.)
> >
> > Thanks for the comment. We have added them to the paper. These parameters can also be found in our source code.

---

### Official Review · AnonReviewer1 · 2020-03-10
**Proposing a post-processing for artery/vain classification**

**Rating:** 1
**Confidence:** 4

**Summary:**

The authors used a cascade U-Net like architecture to segmentation retinal vessels and classified artery and vein. The key idea is to classify the A/V inside of the segmented vessels to exclude the background regions. The proposed post-processing is used to refine the classification results. The authors accomplished a complete segmentation & classification task for retinal vessels.

**Strengths:**

1. An unified segmentation and classification framework for retinal vessels.
2. A/V classification is performed inside the segmented retinal to suppress the imbalance problem.
3. The proposed post-processing procedure efficiently improved the classification accuracy. The post-processing procedure seems to be generally designed, it can be used on other methods theoretically.

**Weaknesses:**

1. The main idea of the proposed SeqNet is to classify the A/V inside of blood vessel region (foreground) to handle the imbalance problem. However, a lot of research has proved that many deep-learning based methods (such as previous work DS-UNet) can handle the imbalance problem between foreground and background quite well. If the imbalance problem is really affect the classification accuracy, the author should give the corresponding comparison experiments.
2. The author should give more detailed validation experiments. Since the improvement of segmentation accuracy is also one major factor affecting the classification accuracy, the authors cannot claim that the improvement is achieved by the joint learning architecture. The authors should rethink their experiments and set baseline properly.
3. I doubt the effectiveness of the joint learning framework. I think the proposed joint learning is equal to a cascade learning framework or maybe even worse. The final loss (CE?) contains the background part (needed for segmentation network), but this background part will still have effect on classification network.  This joint learning framework does not tackle the imbalance issue, theoretically. What about train the classification network with a well-trained segmentation network? Additional comparison experiment is preferred.
4. I think the authors should focus on the post-processing part, since the neural network part confused me and not well explained. Exploring the effectiveness and robustness of simple rule-based post-processing could be interesting (experiments need to be re-designed).

**Detailed Comments:**

1. The illustration of architecture seems too complicated. I think the combination of blue and green modules is one IterNet according to (Li et al., 2019).
2. what is the name 'SeqNet' short for?
3. Reference error. (JIN, 2019) is not correctly referred. Fig. 6 directly follows after Fig. 2.
4. Many typo and grammatical errors should be fixed. Such as:
    Mixed use of 'retinal' and 'retina', incorrect use of word 'etc.', verb tenses mismatch ‘Deep models are also explored and achieved’, 'number of retina images for training is no more than 20' 20??,

**Justification Of Rating:**

This paper aims to propose a novel network architecture to improve the retinal A/V classification accuracy along with a newly proposed post-processing method. However, for the neural network part, both theoretical and experimental proofs are not satisfied. Through the whole paper, this work 'SeqNet' seems emphasize the deep-learning part rather than post-processing. I give strong reject rating.

**Paper Type:**

methodological development

**Questions To Address In The Rebuttal:**

1. More detailed comparison experiments is needed to prove the claims of the proposed network (Weakness 1&2), for example, what about change the backbone (IterNet) with previous works such as DS-UNet or even original U-Net for a baseline.
2. More detailed explanation and experiment is needed to prove the effectiveness of the proposed joint learning framework (Weakness 3). A separate cascade training maybe a simple way to validate the joint learning part.

**Special Issue:**

no

---

> ### Author Response · Authors · 2020-03-27
> **Thanks for the comments.**
>
> Thank you so much for your thoughtful review. We have summarized the changes in a top-level comment called "Summary of Changes in New Version".
>
>
> As for the required additional experiments, we have been trying to redesign some parts of the existing experiments; however, we were not able to conduct all required experiments due to the time limitation. We will keep trying to get the results on data imbalance, segmentation accuracy, etc.
>
> Some other issues:
> 1. The illustration of architecture seems too complicated. I think the combination of blue and green modules is one IterNet according to (Li et al., 2019).
>  -> We have modified the figure and relevant contents according to your comment.
>
> 2. what is the name 'SeqNet' short for?
>  -> It is short for a network sequentially conducting segmentation and classification.
>
> 3. Reference error. (JIN, 2019) is not correctly referred. Fig. 6 directly follows after Fig. 2.
>  -> Has been fixed in this revision.
>
> 4. Many typo and grammatical errors should be fixed. Such as:
>     Mixed use of 'retinal' and 'retina', incorrect use of word 'etc.', verb tenses mismatch ‘Deep models are also explored and achieved’, 'number of retina images for training is no more than 20' 20??,
>  -> We have fixed these errors.
> As for the image number, 20 is correct. Actually, most datasets in this area have very few images. DRIVE, one of the most popular datasets in this area, has 20 images for training and 20 images for testing. LES-AV dataset only has 22 images in total.

---

### Official Review · AnonReviewer3 · 2020-03-11
**Post-processing techniques to ensure vessel segment label consistency**

**Rating:** 3
**Confidence:** 3
**Recommendation:** Poster

**Summary:**

The paper proposed a multi-stage network to perform both vessel segmentation and artery/vein classification. To ensure the label consistency of pixels on the vessel segment, the authors proposed heuristic methods for post-processing. The segmentation network directly borrowed from Li et al., 2019 and the classification is a simple UNet.

**Strengths:**

-The paper is well organized and easy to read
-Figure 4 gives nice demonstration on how the propagation works
-The paper has a nice overview about the recent works
-The main contribution is clearly highlighted


**Weaknesses:**

-Details on how the network got trained are lacking, for example:
	-what's the train/test split?
	-what's the augmentation, optimizer and associated hyperparameters?
-The post-processing method are hand-crafted with a few empirically selected parameters, for example m_A in equation (7). The authors should have discussed how sensitive the post-processing is to these hand picked parameters?


**Justification Of Rating:**

The paper proposed a multi-stage network to perform both vessel segmentation and artery/vein classification. I feel the main novelty lies in the post-processing part where the authors proposed techniques to ensure intra-segment label consistency and inter-segment label propagation

**Paper Type:**

validation/application paper

**Questions To Address In The Rebuttal:**

1. mutli task?-> multistage
2. After equation (6),  … is the angle formed by segments u_i and u_i  -> u_i and u_j
3. After equation (8), … key-point is on the line described by u_{ij} -> w_{ij}?
4. Why propagating artery to vein rather than the other way around?
5. What are the failure cases in the post-processing?

**Special Issue:**

no

---

> ### Author Response · Authors · 2020-03-27
> **Thanks for the comments.**
>
> Thank you so much for your thoughtful review. We have summarized the changes in a top-level comment called "Summary of Changes in New Version".
>
> 1. mutli task?-> multistage
> Yes.
>
> 2. After equation (6),  … is the angle formed by segments u_i and u_i  -> u_i and u_j
> Yes.
>
> 3. After equation (8), … key-point is on the line described by u_{ij} -> w_{ij}?
> Yes. Thanks for all the modifications.
>
> 4. Why propagating artery to vein rather than the other way around?
> They propagate in both ways. Figure 4 is only an illustrative example when propagating artery to vein. We will clarify this in the revision.
>
> 5. What are the failure cases in the post-processing?
> The failures often happen when the vessel structure is very complex. Wrong vessel types may be propagated to neighboring segments, although this erroneous propagation does not affect a lot since the introduction of confidence can suppress excessive propagation.

---

> > ### Author Response · Authors · 2020-04-04
> > **Some other issues**
> >
> > 1. Details on how the network got trained are lacking, for example:
> > -what's the train/test split?
> > -what's the augmentation, optimizer and associated hyperparameters?
> >
> > Train/test splits are 20/20 (official) for DRIVE dataset, and 17/5 (recommended by [1]) for LES-AV.
> > Other details have been introduced in section (d)  of the "Summary of Changes" and have been added into the paper.
> >
> > [1] Hemelings, Ruben, et al. "Artery–vein segmentation in fundus images using a fully convolutional network." Computerized Medical Imaging and Graphics 76 (2019): 101636.
> >
> >
> > 2. The post-processing method are hand-crafted with a few empirically selected parameters, for example m_A in equation (7). The authors should have discussed how sensitive the post-processing is to these hand picked parameters?
> >
> > These parameters are fixed, even for different datasets. We detailed this in section (b)  of the "Summary of Changes".

---

### Official Review · AnonReviewer2 · 2020-03-13
**Interesting sequential approach to Vessel Segmentation+A/V classification, requires quite more work on the evaluation part**

**Rating:** 2
**Confidence:** 5
**Recommendation:** Poster

**Summary:**

The authors propose to solve artery/vein classification in two sequential steps that are trained end-to-end. A first model solves the segmentation task, and then the resulting blood vessel prediction is used as a kind of attention mask on top of the retinal fundus image (by point-wise multiplication) that a second model aimed at performing artery/vein classification uses as its corresponding input. There is a further contribution given by a novel post-processing technique to make predictions more consistent.

**Strengths:**

- The idea of sequentially segmenting and classifying pixels in this specific problem is simple and elegant.
- The post-processing technique seems quite natural and makes sense.
- Results appear to be reasonably good, although I have some reservations (see below).

**Weaknesses:**

- There is lack of very important details in section 2.1. Specifically, I can't find anywhere no mention about the loss function that was minimized. Was there a loss for iternet and a separate loss for the artery/vein branch? Were both sub-networks trained jointly, or IterNet was trained first and only then the artery/vein module was trained afterwards? Learning rate, batch size, optimizer, data preprocessing, and any other technical detail that would allow to reproduce this work are missing in the paper.
- As with any post-processing technique, there are several hidden parameters that need to be tuned by hand. For instance the threshold \theta in eq. (2) to obtain a binary vessel map, or m_A in eq. (7), to name a few. In reality, decomposing a vessel segmentation into segments is a very noisy process and all these parameters may need some tedious adjustment when moving from the training dataset to another dataset. However, the authors avoid this issue in a rather "suspicious" manner: they only use DRIVE and INSPIRE for testing purposes. But since INSPIRE does not have pixel-wise annotations, they just use the segmentation given by IterNet, and therefore the hand-tuned values that they used for DRIVE segmentations will likely be also valid for INSPIRE. I believe we need more experimental evidence on other datasets in order to understand if this really works "universally", see below.
- Results for A/V classification only come in the form of Accuracy. Evaluating this problem is very tricky, because there are some questions left. For instance, what happens to pixels that their method did not find to be vessel pixels, are they included in the accuracy computation for A/V as missed pixels? Or the same question for false-positive vessel pixel, how do we handle them when evaluatiing A/V? Without answering these questions, it is really hard to compare with other people's work. Please see below for some solutions.
- It should be clarified that the method was not retrained on INSPIRE before computing results there (which I hope was the case).
- References are a bit strange, at least the one for IterNet: it does not even mention the journal/conference/arxiv? on which it was published.

**Justification Of Rating:**

The ideas presented in this paper are simple and nice, and I believe there is some novelty in the approach you are proposing. I would be happy to improve my rating if 1) missing technical details are added, and more importantly 2) a more rigorous evaluation of the results, with more datasets and following the steps I indicated above, is reported.

**Paper Type:**

methodological development

**Questions To Address In The Rebuttal:**

A) The above issues may be handled well if the authors were to include more experiments with other datasets. I can point to two of them:
1) HRF has been given A/V labels recently, and you can find them in [3], and they come from [1].
2) There is another nice dataset on which you could test your approach, LES-AV [2].
I believe the correct way to proceed in showing that the approach in this paper works is by testing on these datasets (without retraining!) and reporting results there. Also, hand-tuned parameters in the post-processing stage should not be re-adjusted, because well, if one wants a technique to be useful, it makes little sense to have to re-train or re-adjust things each time we change the data.

[1] Hemelings et al. Artery–vein segmentation in fundus images using a fully convolutional network, 2019.
[2] https://ignaciorlando.github.io/publication/miccai-hemodynamics/
[3] https://github.com/rubenhx/av-segmentation

B) The authors can find also in [3] an evaluation procedure that seems quite rigorous, and it would allow them to compare their technique with the results in [1] in a more "apples to apples" manner. If I remember well, the authors assessed A/V accuracy on "discovered" pixels, correctly segmented vessel pixels. I think they released code in [3] to follow their exact same steps. Unfortunately, and contrary to my point in A), I do think they retrained their system when moving from DRIVE to HRF. In any case, it would still be interesting to see what happens when this technique is compared to [1] without retrained, I believe. If results are much worse, then maybe it would make sense to retrain and report results with and without retraining.

C) Please expand your methods section by filling the technical details that I mentioned in the previous section about the training process.

**Special Issue:**

no

---

> ### Author Response · Authors · 2020-03-27
> **Thanks for the comments.**
>
> Thank you so much for your thoughtful review. We have summarized the changes in a top-level comment called "Summary of Changes in New Version".

---

> > ### Author Response · Authors · 2020-04-04
> > **Other issues.**
> >
> > We think all the issues have been addressed and can be found in the “Summary of Changes”. Here we will give some more descriptions.
> >
> > 1.	There is lack of very important details in section 2.1. Specifically, I can't find anywhere no mention about the loss function that was minimized. Was there a loss for iternet and a separate loss for the artery/vein branch? Were both sub-networks trained jointly, or IterNet was trained first and only then the artery/vein module was trained afterwards? Learning rate, batch size, optimizer, data preprocessing, and any other technical detail that would allow to reproduce this work are missing in the paper.
> >
> > We have added the missing training details in the paper. Please check section (d) of the “Summary of Changes” for the details.
> >
> > 2.	As with any post-processing technique, there are several hidden parameters that need to be tuned by hand. For instance the threshold \theta in eq. (2) to obtain a binary vessel map, or m_A in eq. (7), to name a few. In reality, decomposing a vessel segmentation into segments is a very noisy process and all these parameters may need some tedious adjustment when moving from the training dataset to another dataset. However, the authors avoid this issue in a rather "suspicious" manner: they only use DRIVE and INSPIRE for testing purposes. But since INSPIRE does not have pixel-wise annotations, they just use the segmentation given by IterNet, and therefore the hand-tuned values that they used for DRIVE segmentations will likely be also valid for INSPIRE. I believe we need more experimental evidence on other datasets in order to understand if this really works "universally", see below.
> >
> > We have used the LES-AV as the new evaluation dataset, as your recommendation. Please see section (a) and (c) for the details. However, for the HRF dataset, we are still cannot get its ground-truth labels (no reply after submitting the request form).
> >
> > 3.	Results for A/V classification only come in the form of Accuracy. Evaluating this problem is very tricky, because there are some questions left. For instance, what happens to pixels that their method did not find to be vessel pixels, are they included in the accuracy computation for A/V as missed pixels? Or the same question for false-positive vessel pixel, how do we handle them when evaluatiing A/V? Without answering these questions, it is really hard to compare with other people's work. Please see below for some solutions.
> >
> > We have used the new evaluation metrics. Please refer to section (c) for the results.
> >
> >
> > 4.	It should be clarified that the method was not retrained on INSPIRE before computing results there (which I hope was the case).
> >
> > Now we have abandoned the results from INSPIRE dataset. The results of DRIVE and LES-AV are completely independent without sharing any training materials.
> >
> > 5.	References are a bit strange, at least the one for IterNet: it does not even mention the journal/conference/arxiv? on which it was published.
> >
> > We have fixed the mistakes in the reference part. Thanks again for all the helpful comments. They help improve this paper a lot.

---

### Author Response · Authors · 2020-03-27
**Summary of Changes in New Version**

First, I want to thank all the reviewers for their helpful and valuable comments. We really appreciate them. They have served as outstanding guidance for us to improve the quality of this paper. Here I want to list our responses and what we have done after we got these comments.

(a) About the dataset.
Reviewer 2 advises us to use another two datasets (HRF and LES-AV) to test our approach. We have added the results on the LES-AV dataset in the paper, while we still have no results on the HRF dataset because we cannot get access to its ground-truth A/V labels (submit the request form but get no response at this moment). Also, we have abandoned the results on the INSPIRE dataset because it has no official segmentation labels and makes the evaluation results not convincing enough, as said by Reviewer 2. So in the modified manuscript, we will show the results over the DRIVE and LES-AV datasets. We will add the results of HRF if we get access to the labels in the following weeks. All the results are shown in (c).

(b) Parameters in the post-processing process.
Most parameters in the post-processing process are agnostic to datasets; therefore, the same parameter values can be used for different datasets without manual adjustment. Only one parameter, the maximum distance factor used in Eq. (10), may need to be tuned for the data, since it involves the image size. However, this tuning can be easily avoided by resizing the images into the same or similar sizes. In fact, in our experiments on two different datasets (DRIVE and LES-AV), we do not change any parameter in the post-processing process, demonstrating that it is effective and universal.

(c)  A new evaluation procedure.
A/V classification is a sub-area in this field, but has not drawn much attention from the community, despite its importance. Only a small number of papers have been published in recent years, and few of them have released their source code. This causes some difficulties in performance evaluation. Also, there is no obvious standard in evaluation metrics in this area. However, as “the vast majority of previous approaches are binary classification tech”, a common way is to “compute performance as a binary classification problem, disregarding pixels that do not belong to vessels.”[1] This is also how we conducted evaluations in the initial submission. As pointed out by Reviewer 2, we now know a new evaluation strategy [2] (many thanks to Reviewer 2), we have replaced the former results with the following results. Our method shows obvious advantages on the metric of “discovered centerline pixels of vessels wider than two pixels”, which is the main evaluation metric used in [2].

(c-1) DRIVE dataset

Without post-processing:

Full image
Accuracy: 0.9667042974906049
F1: 0.9665644713287503

Discovered centerline pixels
Accuracy: 0.9137442994181475
F1: 0.9138113536529309

Discovered centerline pixels of vessels wider than two pixels
Accuracy: 0.94596174282678
F1: 0.9459967302987381

Centerline pixels
Accuracy: 0.6909447648492776
F1: 0.7868494402937137

Amount of vessels detected: 0.7735097837640048


With post-processing:

Full image
Accuracy: 0.9670717662746999
F1: 0.9669246028383257

Discovered centerline pixels
Accuracy: 0.9193662525554332
F1: 0.9193995377888029

Discovered centerline pixels of vessels wider than two pixels
Accuracy: 0.953134962805526
F1: 0.9531374281055568

Centerline pixels
Accuracy: 0.695195909388192
F1: 0.7917346885582899

Amount of vessels detected: 0.7777351849593142


Results from [1]:
Full image
Accuracy: 0.965984361740817
F1: 0.9656967838835753

Discovered centerline pixels
Accuracy: 0.8884898439799823
F1: 0.8883101679144095

Discovered centerline pixels of vessels wider than two pixels
Accuracy: 0.9234658153638015
F1: 0.9232625098789554

Centerline pixels
Accuracy: 0.6417887779154821
F1: 0.7448760413197122

Amount of vessels detected: 0.7406059843288463


The result from [2]:
0.9394 (Discovered centerline pixels of vessels wider than two pixels)

---

> ### Author Response · Authors · 2020-03-27
> **continued from above**
>
>
>
> c-2 LES-AV dataset
> This is a relatively new dataset (published in Jan 2018). At this moment, we are able to compare our results only with [1] (the results are directly adopted from their paper because no code for training their models is available now. Will add more comparisons later). The results demonstrate some performance boost by our post-processing as well as a significant performance gain over [1].
>
> Without post-processing:
> Full image
> Accuracy: 0.9775112134804219
> F1: 0.9777353308053521
>
> Discovered centerline pixels
> Accuracy: 0.8584741117257189
> F1: 0.8582065268846948
>
> Discovered centerline pixels of vessels wider than two pixels
> Accuracy: 0.9161816801619433
> F1: 0.9161611953534237
>
> Centerline pixels
> Accuracy: 0.7434082768739482
> F1: 0.7966807777961915
>
> Amount of vessels detected: 0.7757231783229587
>
>
> With post-processing:
> Full image
> Accuracy: 0.9780118802279064
> F1: 0.9782326549666371
>
> Discovered centerline pixels
> Accuracy: 0.8738725270344346
> F1: 0.8736791467424706
>
> Discovered centerline pixels of vessels wider than two pixels
> Accuracy: 0.930161943319838
> F1: 0.9301610076463931
>
> Centerline pixels
> Accuracy: 0.7567427609718207
> F1: 0.8110286991362616
>
> Amount of vessels detected: 0.78461886402309
>
>
>
> The result from [1]: 0.86 (accuracy of A/V classification on vessel pixels)
>
> (d) The implementation details.
> Some reviewers ask us to provide the technical details including loss, learning rate, etc. We did not give these details in our initial submission because there is a strict page limit and we think this information can be easily found in the source code, which will be made public later. In this revision, we have added some basic descriptions to help reproduce our method. For example, “Both two steams use separate cross-entropy losses and are trained jointly with a batch size of 16. As the target, IterNet uses the segmentation labels while the classification part uses the A/V labels. Adam is used as the optimizer with the learning rate of 0.001.” (Section 2.1)
>
> (e) Typos, grammar mistakes, and format errors (especially in the references).
> Thanks for kindly reminding us about the mistakes. In the revised version, we fix all of them.
>
>
> For other issues, please check our responses for respective reviewers’ comments.
>
> [1] Galdran, Adrian, et al. "Uncertainty-Aware Artery/Vein Classification on Retinal Images." 2019 IEEE 16th International Symposium on Biomedical Imaging (ISBI 2019). IEEE, 2019.
> [2] Hemelings, Ruben, et al. "Artery–vein segmentation in fundus images using a fully convolutional network." Computerized Medical Imaging and Graphics 76 (2019): 101636.

---

### Meta-Review · Area_Chair1 · 2020-04-10
**MetaReview of Paper163 by AreaChair1**

**Rating:** 3
**Recommendation For Accepted Papers:** Poster

**Metareview:**

The manuscript introduced a unified segmentation and classification framework for retinal vessels that is also able to do A/V separation. Two authors are supportive while a  third is less so. However, some of the critiques by the unsupportive reviewers are less argued and justified than the positives of supporting reviewers. In addition, the authors have done a fair amount of very good work in addressing the concerns of the reviewers. They undertook additional experiments and adddressed each of the reviewers's points in turn and improving the actual manuscript. I am supportive of the acceptance and  believe this is a paper somewhere between a 3 and a 4.

**Paper Type:**

methodological development

**Special Issue:**

no

---

### Decision · Program_Chairs · 2020-04-11

Accept